

# Accuracy of deep learning, a machine learning technology, using ultra-wide-field fundus ophthalmoscopy for detecting idiopathic macular holes

Toshihiko Nagasawa[1,*], Hitoshi Tabuchi[1,*], Hiroki Masumoto[1], Hiroki Enno[2], Masanori Niki[3], Hideharu Ohsugi[1] and Yoshinori Mitamura[3]

[1] Department of Ophthalmology, Tsukazaki Hospital, Himeji City, Hyogo Prefecture, Japan
[2] Rist Inc., Tokyo, Japan
[3] Department of Ophthalmology, Institute of Biomedical Sciences, Tokushima University, Tokushima City, Tokushima Prefecture, Japan
[*] These authors contributed equally to this work.

## ABSTRACT

We aimed to investigate the detection of idiopathic macular holes (MHs) using ultra-wide-field fundus images (Optos) with deep learning, which is a machine learning technology. The study included 910 Optos color images (715 normal images, 195 MH images). Of these 910 images, 637 were learning images (501 normal images, 136 MH images) and 273 were test images (214 normal images and 59 MH images). We conducted training with a deep convolutional neural network (CNN) using the images and constructed a deep-learning model. The CNN exhibited high sensitivity of 100% (95% confidence interval CI [93.5–100%]) and high specificity of 99.5% (95% CI [97.1–99.9%]). The area under the curve was 0.9993 (95% CI [0.9993–0.9994]). Our findings suggest that MHs could be diagnosed using an approach involving wide angle camera images and deep learning.

## INTRODUCTION

In 1988, Gass described idiopathic macular holes (MHs) as a retinal break commonly involving the fovea (*Gass, 1988*), and in 1991 Kelly and Wendel reported that MHs can be successfully repaired through vitreous surgery (*Kelly & Wendel, 1991*). The age and gender adjusted annual incidences of primary MH have been reported at 7.9 eyes and 7.4 respectively per 100,000 inhabitants, and the male to female ratio was 1:2.2 (*Forsaa et al., 2017*). The accepted pathogenesis has macular hole formation proceeding in stages from an impending hole to a full thickness MH, with visual acuity deteriorating to less than 6/60 in 85% of cases (*Luckie & Heriot, 1995*). The development of optical coherence tomography (OCT) and improvement of image resolution have made the diagnosis of macular diseases substantially easy (*Kishi & Takahashi, 2000*).

Corresponding author
Toshihiko Nagasawa,
t.nagasawa@tsukazaki-eye.net

In addition, the advent of wide angle fundus cameras has made the observation of the entire retina possible through a simple and noninvasive approach (*Nagiel et al., 2016*). An example of such a camera is the ultra-wide-field scanning laser ophthalmoscope (Optos 200 Tx; Optos PLC, Dunfermline, United Kingdom), which is known as Optos. It is capable of photographing the fundus without mydriasis, and it is used for making judgments regarding the diagnosis, follow-up, and treatment effects of various fundus diseases (*Prasad et al., 2010*; *Wessel et al., 2012*; *Ogura et al., 2014*). Optos can minimize the risk of a rise in pupillary block caused by mydriasis and intraocular pressure increase. This makes Optos suitable for medical use in remote areas where the services of ophthalmologists are limited, as the device can be safely used by orthoptists and other medical professionals.

Recently, image processing technology applying deep learning, a sub-field of machine learning algorithm studies, has attracted attention because of its very high classification performance. The use of this technology for medical images is being actively studied (*LeCun, Bengio & Hinton, 2015*; *Liu et al., 2015*; *Litjens et al., 2016*). In the ophthalmic field, there are reports on the use of the ocular fundus camera and deep learning and on the improvement in the accuracy of automatic diagnosis of diabetic retinopathy and retinal detachment with these approaches (*Gulshan et al., 2016*; *Ohsugi et al., 2017*; *Ryan et al., 2018*). However, the diagnostic accuracy of the wide angle ocular fundus camera for macular diseases is yet to be reported. Deep neural networks have been used to diagnose skin cancer with as much accuracy as that attained by dermatologists (*Esteva et al., 2017*). We decided to assess the diagnostic capability of deep neural networks for macular holes as compared with ophthalmologists' diagnoses.

The present study assessed the presence of MHs, which are considered as a macular disease, using ultra-wide-field fundus images with deep learning in order to determine the accuracy of deep learning, and to compare the ophthalmologist and the deep neural network for MHs.

## MATERIALS AND METHODS

### Data set

The study dataset included 910 Optos color images obtained at the Tsukazaki Hospital (Himeji, Japan) and Tokushima University Hospital (715 normal images and 195 MH images). Of the 910 images, 637 were used for training purposes (80%; 501 normal images and 136 MH images; learning images) and 273 were used for testing purposes (20%; 214 normal images and 59 MH images; test images).

The 637 learning images underwent image processing and were amplified to 5,000 images (3,887 normal images and 1,113 MH images). The image amplification process comprised contrast adjustment, $\gamma$ correction, histogram equalization, noise addition, and inversion. We performed training on these learning images with a deep convolutional neural network (CNN) and constructed a deep learning model.

Cases of MHs were confirmed by a retinal specialist who conducted fundus examinations using an ophthalmoscope and OCT. For OCT, a swept-source OCT system (SS-OCT; DRI OCT-1 Atlantis, TOPCON Corporation, Tokyo, Japan) was used. All Optos images

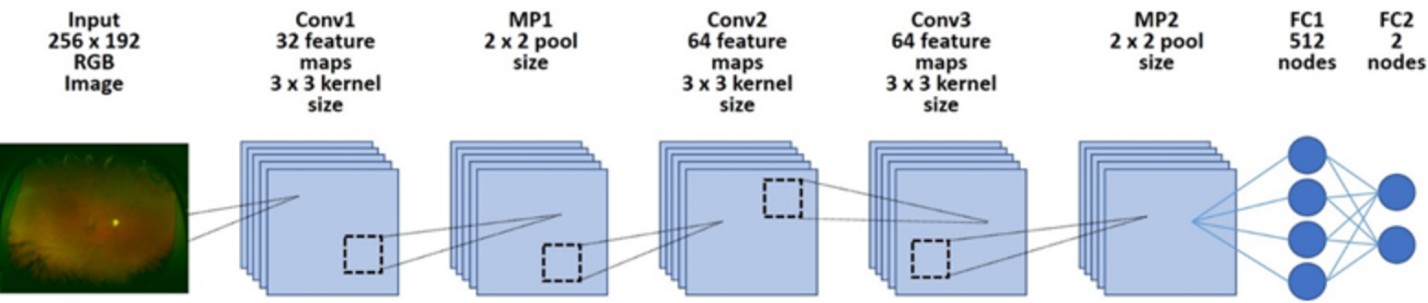

**Figure 1  Overall architecture of the deep learning model.** First, each dataset's image was reduced to 256 × 192 and was input into the model. Next, it was passed through all convolution layers and the entire binding layer, and it was classified into two classes.

obtained from the MH patient database were considered for inclusion. Images from patients complications, such as vitreous hemorrhage, asteroid hyalosis, intense cataract, and retinal photocoagulation scars, and other conditions, such as fundus diseases, were excluded. Additionally, images with poor clarity were excluded. Moreover, images from patients with stage 1 MHs (according to the classification by *Gass, 1995*) and those with retinal detachment were excluded.

The procedures used conformed to the tenets of the Declaration of Helsinki, and an informed consent was obtained from either the subjects or their legal guardians after explanation of the nature and possible consequences of the study. An approval was obtained from the Institutional Review Board of Tsukazaki Hospital (No 171001) and Tokushima University Hospital (No 3079) to perform this study.

## Deep learning model

We implemented a deep learning model using a CNN (Fig. 1). We arranged three convolutional layers. The rectified linear unit (ReLU) activation function and batch normalization were placed after each convolutional layer. A max pooling layer (MP 1, 2) was placed after convolutional layers 1 and 3. In addition, a dropout layer (drop rate 0.25) was placed after each max pooling layer (MP 1, 2). Finally, the two fully connected layers (FC 1, 2) were arranged and classified into two classes using the Softmax function.

## Training the deep convolutional neural network

All obtained image data were converted to 256 × 192 pixels. Learning was carried out with mini-batch processing of 10 images and an epoch number of 100. The initial value of the network weight was randomly provided as the zero average of Gaussian distribution, with a standard deviation of 0.05. Dropout processing was performed to mask the first total tie layer (FC1), with 50% probability. The network weights were optimized using stochastic gradient descent (SGD) with momentum (learning coefficient, 0.01; inertia term, 0.9). Of 100 deep learning models obtained in 100 learning cycles, the model with the highest accuracy rate for the test data was selected as the deep learning model.

## Outcome

The area under the curve (AUC) and sensitivity/specificity were determined for the ability of the selected CNN model to discriminate between normal eyes and MH.

## Statistical analysis

The receiver operating characteristic curve (ROC curve) and the 95% confidence interval (CI) of the AUC were obtained. The ROC curve was created by considering that the value judged to involve MHs exceeded the threshold (cutoff value) as positive. The model was fitted to only 90% of the test data, and 10% were thinned out. We created 100 ROC curves by making 100 patterns. One hundred AUCs were calculated from the ROC curves. With regard to the AUCs, 95% CI were obtained by assuming normal distribution and using standard deviation. With regard to sensitivity and specificity, the first of the 100 ROC curves were used, and the sensitivity and specificity at the optimum cutoff value calculated using Youden Index 23 as the representative value of the deep learning model were used. The accuracy, specificity, sensitivity, and response times by CNN and six ophthalmologists were calculated.

## Creation of an ophthalmologist application

Of the 273 test images, 50 normal images and 50 MH images were extracted using the random number generation method (equal representation for normal data and the disease data). We calculated the accuracy, specificity, sensitivity, and response times by CNN based on the averaged results of six ophthalmologists.

## Determination and measurement methods for calculating the required time

Six ophthalmologists determined the presence or absence of MHs in 50 images presented on a computer monitor. The answer inputs of either 0 or 1 on the response form were populated in an Excel table.

The time taken by the ophthalmologists to enter data in the computer was also included. In deep neural network, a series of tasks was performed for all presented numbers as follows: confirming the number of the problem in the answer column → reading the image → judging → filling in the answer column. The total time was counted as the operation time. This series of work was performed 15 times by a computer, and the working time was considered as the median value. The time required by the ophthalmologists was set as the time taken to complete all answers in the Excel file. The time required for the deep neural network was measured by the internal clock of the computer. The specifications of the computer were as follows: operating system, Windows 10 Home; CPU, Intel Core i7 - 3630 QM; memory, 8.00 GB; GPU, NA.

## Heat map

Using the gradient weighted class activation mapping (Grad-CAM) (*Selvaraju et al., 2016*) method, we obtained a heat map of the coordinate axes in the image focused on by the CNN. The layer that used the gradient was specified as convolution layer 2. Additionally, we specified ReLU as the backprop modifier.

**Table 1  Demographic data.** No statistically significant differences were observed between the groups. Data are presented as numbers (%) unless otherwise indicated.

|  | Macular hole images | Normal images | *p*-value |  |
|---|---|---|---|---|
| *n* | 195 | 715 |  |  |
| Age | 66.9 ± 7.6 (20∼85) | 67.3 ± 12.2 (11∼94) | 0.5726 | Student's *t*-test |
| Sex (female) | 117 (60%) | 390 (54.6%) | 0.1933 | Fisher's exact test |
| Eye (left) | 102 (52.3%) | 361 (50.5%) | 0.6865 | Fisher's exact test |

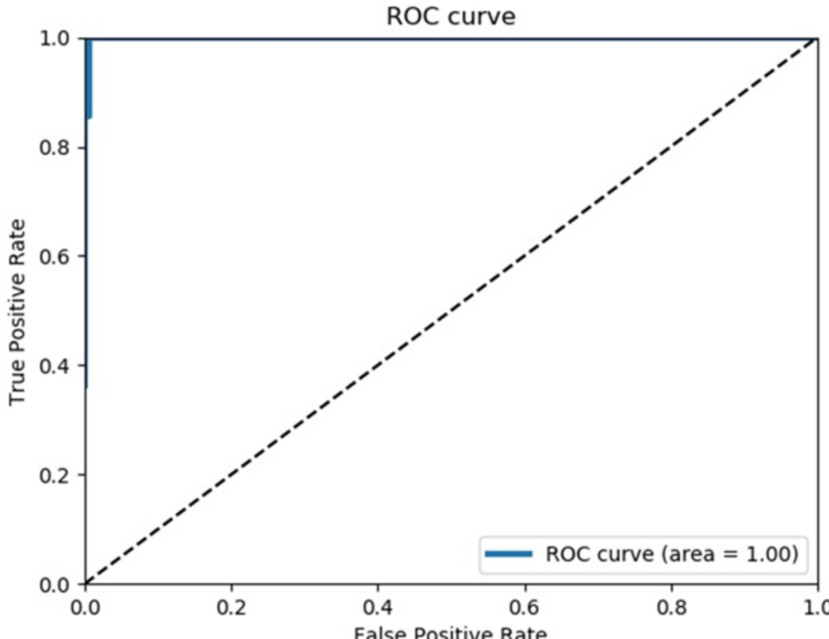

**Figure 2  Receiver operating characteristics curve.** This is the first one out of 100 ROC curves. The average AUC of 100 ROC curves was almost 1, and all ROC curves were similar.

## RESULTS

### Background data

Table 1 shows the total number of normal and MH images, patient age, patient sex, and left/right of the imaged eyes. There were no statistically significant differences between the normal and MH images with regard to age, sex ratio, and left eye ratio (Student's *t*-test and Fisher's exact test).

### Evaluation of the performance model

The mean value of 100 AUCs prepared by the CNN model was 0.9993 (95% CI [0.9993–0.9994]).

The first curve among the 100 calculated ROC curves is shown in Fig. 2.

The mean sensitivity obtained from the 100 ROC curves was 100% (95% CI [93.5–100%]), and the mean specificity was 99.5% (95% CI [97.1–99.99]%).

**Table 2  The results of CNN model and overall ophthalmologist.** The convolutional neural network model, discrimination test of the macular holes data and the normal data, ophthalmologist, accuracy, sensitivity, specificity, and measurement time.

|  | CNN model | Overall Ophthalmologist |
|---|---|---|
| Accuracy | 100% | 80.6 ± 5.9% |
| Specificity | 100% | 95.2 ± 4.3% |
| Sensitivity | 100% | 69.5 ± 15.7% |
| Measurement time (s) | 32.80 ± 7.36 | 838.00 ± 199.16 |

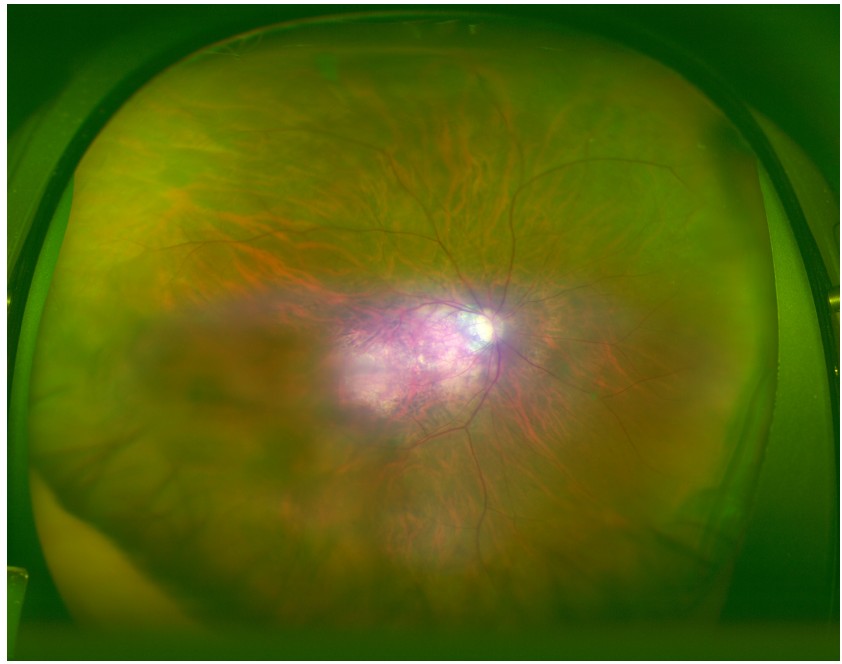

**Figure 3  Heatmap superimposed on the photo.** The dark blue color shows the point where the deep neural network is paying attention on the macula and from the same point of view of an ophthalmologist.

Ophthalmologists carried out the test, and the mean (standard deviation) required time was 838.00 s (±199.16), the mean (standard deviation) accuracy rate was 80.6% (5.9%), sensitivity was 65.9% (15.7%), and specificity was 95.2% (4.3%). The same test was carried out with the CNN model, and the mean (standard deviation) required time was 32.8 s (±7.36) and accuracy rate, sensitivity, and specificity were all 100% (Table 2).

### Heat map

An image with the corresponding heat map superimposed was created by the CNN, and the focused coordinate axes in the image were indicated. A representative image is presented in Fig. 3. Focal points accumulated on the heat map at the fovea of the fundus macula. It is suggested that the CNN may distinguish s diseased eye from a normal eye by focusing on the MH lesion site.

Blue color was used to indicate the strength of CNN attention. The color became stronger on one side of the arcade, with centering at the macular fovea, and accumulation was noted at the focus points.

## DISCUSSION

OCT is considered indispensable for the diagnosis of MHs. However, in the present study, MHs were diagnosed using images from a wide angle camera and deep learning. Optos adopts the method of combining a red (633 nm) laser image and a green (532 nm) laser image to give a false color. Details of color information are inferior to those of a conventional fundus camera. Therefore, the quality of the diagnosis made by an ophthalmologist might reduce. With the deep learning model, the approach is different from the approach of an ophthalmologist, with a focus only on the difference from a normal eye, and there is a possibility that some additional general and flexible features of learning can be considered. The heat map spreads over a relatively wide area around the macula fovea, and this approach appears to have a classification that is superior to the judgment ability of an ophthalmologist.

The present study has several limitations. When light transmission in the eye is absent because of intense cataract or dark vitreous hemorrhage, it is difficult to obtain images with Optos, and such cases were not included in the present study. In addition, this study only compared normal eyes and MH eyes, and it did not assess eyes affected by other fundus diseases. This warrants the preparation of a large scale data set for applying deep learning. Although the diagnostic ability of using a wide angle ocular fundus camera and deep learning for diabetic retinopathy and retinal detachment has been reported, the findings of this study indicate the high diagnostic ability of this approach for MHs, which are considered a macular disease. In the future, studies should assess the possibility of performing automatic diagnoses with a wide angle camera for other macular diseases, such as macular epiretinal membrane and age-related macular degeneration.

If Optos is used in a medically depopulated area, wide-area ocular fundus photography can easily be performed under a non-mydriasis condition, without medical complications. Moreover, even if no ophthalmologist is available to assess the image, the deep-learning algorithm can be used for MH diagnosis, as it has a high accuracy rate for MH diagnosis. Many regions of the world have an inadequate number of ophthalmologists (*Resnikoff et al., 2012*) and thus, the automatic diagnosis of MH using Optos fundus images has great potential. If surgical treatment is performed at an appropriate time in MH patients, a good prognosis can be obtained. The results of this study strongly support the use of an Optos based telemedicine system. Such systems might aid in the early detection of patients with MHs in areas where ophthalmologists are absent.

## CONCLUSIONS

Using ultra-wide-field fundus images, deep learning, could successfully diagnose MHs. We believe that this approach will be very useful in the practical clinical diagnosis of MHs.

Further research with increasing number of sheets, deepening the layer structure, and using metastasis learning are necessary to confirm our results.

## ACKNOWLEDGEMENTS

The authors would like to thank Enago (http://www.enago.jp) for the English language review.

### Funding

The authors received no funding for this work.

### Competing Interests

Hiroki Endo is employed by Rist Inc., Tokyo, Japan.

### Author Contributions

- Toshihiko Nagasawa analyzed the data, contributed reagents/materials/analysis tools, approved the final draft.
- Hitoshi Tabuchi conceived and designed the experiments, analyzed the data, prepared figures and/or tables.
- Hiroki Masumoto conceived and designed the experiments, performed the experiments, prepared figures and/or tables.
- Hiroki Enno conceived and designed the experiments, performed the experiments.
- Masanori Niki and Hideharu Ohsugi contributed reagents/materials/analysis tools.
- Yoshinori Mitamura contributed reagents/materials/analysis tools, authored or reviewed drafts of the paper.

### Clinical Trial Ethics

The following information was supplied relating to ethical approvals (i.e., approving body and any reference numbers):

An approval was obtained from the Institutional Review Board of Tsukazaki Hospital (No 171001) and Tokushima University Hospital (No 3079) to perform this study.

### Data Availability

miki, masayuki (2018): MH_heatmap_mask_re_upload. figshare. Fileset. https://doi.org/10.6084/m9.figshare.7133075.v2.

miki, masayuki (2018): Normal_heatmap. figshare. Fileset. https://doi.org/10.6084/m9.figshare.6742163.v1.

miki, masayuki (2018): MH_heatmap_nomask. figshare. Fileset. https://doi.org/10.6084/m9.figshare.6742166.v1.

### Clinical Trial Registration

The following information was supplied regarding Clinical Trial registration: 171001, 3079.

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
