# Peer review of "Accuracy of deep learning, a machine learning technology, using ultra-wide-field fundus ophthalmoscopy for detecting idiopathic macular holes"

_PeerJ, doi:10.7717/peerj.5696_

## Round 0.1 · original submission · Major Revisions

Thanks you for submitting your article to PeerJ. It is an interesting article. However, a number of important items were raised during by the reviewers, most importantly by Reviewer 4. Please address these.

Reviewer 1 ·

Basic reporting

1. line 61-
Please correct Optus to Optos and rewrite the sentence as “The study dataset included 910 Optos color images obtained at Tsukazaki Hospital (Himeji, Japan) and Tokushima University Hospital (715 normal images and 195 MH images).” because the original one was difficult to read.

2. line 111, ROV should be ROC.

3. line 125-
I don’t know what this sentence meant.

4. line 203
The authors wrote that "If surgical treatment is performed at an appropriate time in MH patients, a good prognosis can be obtained".
How the Optos-based telemedicine system is used for the determination of appropriate timing?

Experimental design

1. As the authors commented, the limitation of this study was the inclusion of only normal and MH eyes.

2. line 81-,
When did authors obtain the informed consent from each subject? Were all images used in this study collected for the purpose of this study after obtaining the informed consent from each subject? According to the clinical research ethical guidelines, the researchers can include the existent data after they disclose the research information.

Validity of the findings

Although I admit the accuracy of the AI, the scores of ophthalmologists for the diagnosis of MH were low, especially the sensitivity. Were those ophthalmologists instructed 1:1 ratio of the normal and the MH image in the data set?

Additional comments

1. There are eyes having “pseudo” MH. Please discuss whether the AI can differentiate true and pseudo MH.

2. The results discourage the "real" ophthalmologists. In addition to the speed, the accuracy of the diagnosis was superior in the AI than in the ophthalmologists. Please discuss the role of the ophthalmologists in the future.

Reviewer 2 ·

Basic reporting

Good enough.

Experimental design

Concise and good.

Validity of the findings

Data is robust, statistically sound.

Additional comments

Nagasawa et al. reported the possibility of deep learning for the detection of idiopathic macular holes in ultra-wide-field fundus images. This study is unique and new. I think this paper has a sufficient priority to be published in PeerJ. However, I also have several minor concerns in this manuscript.

1. line 39; the authors emphasize that Optos dose not need mydriasis. In the current study, it is not clear all the Optos images were taken under the condition of non-mydriasis.
2. line 75; Images from patients, complications, such as vitreous hemorrhage, asteroid hyalosis, 76 intense cataract, and retinal photocoagulation scars, and other conditions, such as fundus diseases, were excluded. Additionally, images with poor clarity were excluded. Moreover, images from patients with stage 1 MHs and those with retinal detachment were excluded. The authors need to describe how many Optus images were excluded from all images.
3. Table 2; it is unclear what 32:80±7:36 and 13:58:00±3:19:16 actually mean.
4. I am not sure why the authors use Optos to detect MH. OCT should be more accurate, easy, and more common.

Reviewer 3 ·

Basic reporting

In the table, it is not clear what format and units the time is reported in.

The figure legends should allow the figure to be read without referring to the original article – they may need to be made slightly more descriptive.

I’m not sure ROC curve is necessary or helpful when the AUC is essentially 1.

The image preprocessing is not well described. The images appear to have a circular crop applied to the original image – this should be described.

Was any effort made to center the images, align the disc, or flip left/right eyes to make the images appear similar to the CNN?

Experimental design

no comment

Validity of the findings

no comment

Additional comments

From an image processing perspective, assuming a good quality fundus image, the detection of a macular hole (a small dark circle in a larger fairly homogenous image) is not that complicated, and thus it is not surprising the CNN works as well as it does, but the results are nonetheless impressive.

From a clinical utility perspective, it is not clear that this is a solution to an existing clinical problem since macular holes always cause visual loss in the stages included in this study, the rationale for creating a screening program to detect them is less compelling. While perhaps not necessary for publication here, it may strengthen the paper to add some discussion as to how such a program might be used in the real world.

Reviewer 4 ·

Basic reporting

Introduction
For me, the intro is far too short and doesn’t really describe the problem in enough detail. There is almost no clinical background, and the discussion around deep learning is too brief. I would suggest expanding the Introduction to cover the following topics:

- What is a macular hole? How does it appear in a fundus photo vs. OCT?
- What is the prevalence of macular holes? Some statistics might be helpful
- What are the complications associated with MH, if left undiagnosed?
- Deep learning is not a machine learning algorithm; it’s a sub-field of research within ML
- You state that DL is good generally, but you should give details of why DL is a good approach specifically to your problem. Have other methods been tried previously for MH? Are they inadequate?
- Please cite some other recent DL papers in the context of ophthalmology, especially this one: https://www.nature.com/articles/s41551-018-0195-0

Methods

- It probably makes more sense to describe the FC dropout layer in the section “Deep learning model”, rather than the “Training…” section.
- Lines 132-135 do not make sense - please revise these sentences to be more clear
- A citation is needed for the Grad-CAM method on line 141
- What cost function did you use? Cross-entropy, or something else?

Results

I am quite confused about the methods on page 9. Specifically, how (or why) is “deep-learning response time” calculated by the ophthalmologists (line 125)? The description on line 132-135 about data entry is also unclear, particularly the sentence: “In deep learning, a series of tasks was performed for all presented numbers as follows…”. My best guess is that the authors are trying to fairly compare the DL computation time with the ophthalmologists’ time taken to record the same information. Please revise this section to be more clear.

Regarding the figures, I think there a few things that can be improved:

- In my opinion, the legends are too short. I personally try to provide enough information in the figures so that a reader could get the gist of the whole paper by reading the legends alone.
- For Figure 2, I’d suggest zooming in on the ROC curve figure, perhaps with the x and y axes at 0.5 or something. You really can’t make anything out otherwise. I’d suggest also including curves from several runs - perhaps the best, worst and average? It’ll give readers a better sense of the variability.
- Figure 3 showing the heat map is not all that informative without a colorbar. It also might be useful to include a few examples rather than just one.
- Table 2: “Accuracy” is a better term for “correct answer rate”. Also please state the unit of measurement time

Other points:

- What is the “first” curve? The first experiment you ran? Why not the best curve?
- On line 162, is this 13 minutes per image?

Conclusions

- There’s no need to repeat that deep learning is an ML technology.
- What are you going to do next?

Grammar, spelling and formatting
Overall the language is very good, though there are a few spelling/grammatical errors:
- Missing space after “macular holes” (line 19)
- Optus → Optos (line 61)
- Lots of unnecessary hyphens in the terms deep-learning and machine-learning (line 71, 88 and various other places)
- “...using a CNN” (line 88)
- “The rectified linear unit (ReLU) activation function…” (line 89)
- What is meant by a ‘tie layer’? Not sure what this means (line 92, 100)
- “The network weights were optimized using stochastic gradient descent (SGD) with momentum…“ (line 101-102)
- ROV → ROC, and various grammatical errors afterward (line 111 onwards)
- Background data (line 147)
- Probably better to describe the eye in terms of left/right or OD/OS (line 148)
- Resions → regions (line 201)

Experimental design

The research question is not all that well defined in the introduction. Ultimately, the goal was to evaluate the performance of DL algorithm for detecting MH. However, the authors also do a good job of comparing the algorithm to multiple experts; something many papers do not do. I would therefore suggest adding a couple of sentences at the end of the introduction to state that this was also part of the study.

My main issue with the overall experimental design relates to how the final model for evaluation was selected. You shouldn’t use test accuracy as the basis, but instead use a validation set. Later, in the “Statistical analysis” section, I don’t really understand the authors’ description of the ROC analysis. I get that there should be one curve per model (100 overall) but I do not understand what is meant by: “We created 100 ROC curves by making 100 patterns, and 10% were thinned out”. Some clarification is needed. The authors also state the model was fitted to only 90% of the test data. Presumably this is an error, and the authors mean training data. This would suggest that the authors did indeed use a 10% validation set, but this is unclear. Please revise this section to better describe how the model was tested and evaluated.

Some other points:

- It’s very common to pre-train networks on bigger datasets to boost performance and reduce the time needed to train. Did you try this out?
- Images from patients with various complications were excluded. How many? Why?
- What was the criteria for “clarity”?
- What is “stage 1” MH and why are they not included?
- You state that 100 models were trained - what was the variability over the 100 runs?

Validity of the findings

As described above, I am concerned about how the final model performance was evaluated. Running the algorithm 100 times and picking the best one based on testing accuracy is cheating a little bit, although it’s not clear from the methods whether this is how it was actually done. Furthermore, given the size of the dataset and near-perfect performance on the test set, I think that cross-validation is necessary to get true sense of algorithm generalizability. I think that given the authors can afford to do 100 repeats on the same data, performing K-fold cross validation should be feasible and would strengthen the reporting.

Given that the authors went to the effort of making an application to capture six experts’ gradings, it would be good if the authors could report metrics of interrater agreement (e.g. kappa coefficients). Furthermore, it’s important that the authors discuss the limitations of the reference standard used, especially if it was based on the diagnosis of a single grader. This paper gives some good insight into why this is important: https://research.google.com/pubs/pub46802.html

Additional comments

Overall, the paper is well written and demonstrates that DL is a powerful approach for classifying MH in wide-angle retinal photographs. The choice of metric is appropriate and the comparison with multiple raters is great to see. Some more background information about MH would be useful for readers outside of the discipline, and would help to contextualize why this work is important.

Ultimately, I am concerned that the results do not reflect the realistic real-world performance of the proposed method. With further clarification of how the model was tested and evaluated, I think that most of concerns will be addressed. However, I would maintain that cross-validation would be a more appropriate way of assessing the generalizabiltiy of the CNN.

---

## Round 0.2 · Minor Revisions

Thank you for addressing many of the issues raised by the reviewers. However, a few minor issues remain as detailed.

Reviewer 4 ·

Basic reporting

The response to my question "What are the complications associated with MH, if left undiagnosed?" is missing the point somewhat. It might well be obvious in the field of ophthalmology... but this is not an ophthalmology journal. Non-experts should come away knowing *why* this work is important. Not a massive issue, but a tad dismissive in my opinion.

I also still have issue with the "Statistical analysis" section. This sentence makes no sense to me: "The model was fitted to only 90% of the test data. We created 100 ROC curves by making 100 patterns, and 10% were thinned out." You should not have fitted *anything* to the test data, please correct this typo or clarify what you mean.

"Correct answer rate" is still used in several places - to me this should be "accuracy".

Fully connection layers --> Fully connected layers

The remaining comments have been addressed satisfactorily.

Experimental design

The comments have been addressed satisfactorily.

Validity of the findings

The comments have been addressed satisfactorily.

Additional comments

Thank you for addressing the comments from the original review. With the exception of a few lingering issues from the original (and a few typos here and there) I think that it merits publication.

---

## Author Rebuttal · Round 0.2

1 **Toshihiko Nagasawa, MD, Department of Ophthalmology, Saneikai Tsukazaki Hospital,**

2 **HiJapan, 68-1 Aboshi Waku, Himeji City, Hyogo Prefecture 671-1227, Japan**

3 **E-mail: t.nagasawa@tsukazaki-eye.net**

4 **Phone: +81 79-272-8555; Fax: +81 79-272-8550**

6 **Jayashree Kalpathy-Cramer**

7 **Editor**

8 *PeerJ*

9 **Subject: Manuscript article ID: 26404**

11 Dear Jayashree Kalpathy-Cramer and Reviewers

13   Thank you for the thoughtful and constructive feedback you provided regarding our

14 manuscript article ID: 26404, entitled "Accuracy of deep learning, a machine-learning

15 technology, using ultra–wide-field fundus ophthalmoscopy for detecting idiopathic macular

16 holes."

17   We are thankful for all your suggestions to improve our paper, and we have revised it

18 accordingly and formatted it to conform to the PeerJ guidelines.

19   Please find enclosed detailed responses to each one of the reviewers' comments. We have

20 aimed to explain our rationale in each case and hope our modifications and clarifications will

21 render our revised manuscript suitable for publication.

22   We all agree to the revised version of our manuscript and hereby resubmit it for a secondary

23 evaluation. Thank you once again for your

24   consideration of our paper.

25 Sincerely,

26

27

**Reviewer 1 (Anonymous)**

Basic reporting

1. line 61-

Please correct Optus to Optos and rewrite the sentence as "The study dataset included 910 Optos color images obtained at Tsukazaki Hospital (Himeji, Japan) and Tokushima University Hospital (715 normal images and 195 MH images)." because the original one was difficult to read.

Thank you for suggestion, we have changed "The study dataset included 910 Optos color images obtained at the Tsukazaki Hospital (Himeji, Japan) and Tokushima University Hospital (715 normal images and 195 MH images)."

2. line 111, ROV should be ROC.

Thank you for pointing this error out, we have changed "ROV→ROC"

3. line 125-

I don't know what this sentence meant.

Thank you for pointing this out, we have changed our sentence: "We calculated the correct answer rate, specificity, sensitivity, and response times by CNN and six ophthalmologists were calculated."

4. line 203

The authors wrote that "If surgical treatment is performed at an appropriate time in MH patients, a good prognosis can be obtained". How the Optos-based telemedicine system is used for the

51    determination of appropriate timing?

52    This is an interesting perspective. First of all, we believe that this system can be used in areas

53    without access to an ophthalmologist. There is a potential for diagnoses to be made by

54    optometrists to be able to reach patients in remote resions that might otherwise be missed.

55

56    Experimental design

57    1.  As the authors commented, the limitation of this study was the inclusion of only normal and

58        MH eyes.

59    You have raised an important issue. It is true that we included only normal and macular hole

60    images for this proof-of-concept study, but in the future we want to experiment to assess

61    whether more comprehensive diagnoses can be accomplished as well.

62

63    2. line 81-,

64    When did authors obtain the informed consent from each subject? Were all images used in this

65    study collected for the purpose of this study after obtaining the informed consent from each

66    subject? According to the clinical research ethical guidelines, the researchers can include the

67    existent data after they disclose the research information.

68    Thank you for observation. We obtained consents forms from patients after explaining the

69    research information to them. This information can be found in the methods section of the

70    revised manuscript.

71

72    Validity of the findings

73    Although I admit the accuracy of the AI, the scores of ophthalmologists for the diagnosis of

74    MH were low, especially the sensitivity. Were those ophthalmologists instructed 1:1 ratio of the

75    normal and the MH image in the data set?

76    You have raised an important issue. We did not inform the ophthalmologists of the 1:1 ratio,

77    as AI does not know about it either. We think this was a fair experimental setting.

78

79    Comments for the Author

80    1.   There are eyes having "pseudo" MH. Please discuss whether the AI can differentiate true

81       and pseudo MH.

82    Thank you for observation. In this study, we did not investigate the diagnosis of pseudo MH,

83    because the number of cases is considered low, and the condition was not suitable for our deep

84    learning study. We do not know whether the AI can differentiate true and pseudo MH at the

85    moment.

86

87    2. The results discourage the "real" ophthalmologists. In addition to the speed, the accuracy of

88    the diagnosis was superior in the AI than in the ophthalmologists. Please discuss the role of the

89    ophthalmologists in the future.

90    Thank you for your commentary. The six ophthalmologists are in the study were not retinal

91    specialists, and they diagnosed MH only from looking at the Optos images. In the actual clinical

92    practice, ophthalmologist make a more comprehensive diagnosis, their expertise is invaluable,

93    and they will still be needed in the near future.

94

95

96

97    **Reviewer 2 (Anonymous)**

98    Basic reporting

99

100    Comments for the Author

101   1.  line 39; the authors emphasize that Optos dose not need mydriasis. In the current study, it

102      is not clear all the Optos images were taken under the condition of non-mydriasis.

103   Thank you for pointing this out. We have explained that mydriasis is not required to obtain

104   appropriate Optos images in general. However, we did not differentiate images taken under

105   mydriasis in our study because the cases were mixed.

106

107   2.  line 75; Images from patients, complications, such as vitreous hemorrhage, asteroid hyalosis,

108      76 intense cataract, and retinal photocoagulation scars, and other conditions, such as fundus

109      diseases, were excluded. Additionally, images with poor clarity were excluded. Moreover,

110      images from patients with stage 1 MHs and those with retinal detachment were excluded.

111      The authors need to describe how many Optus images were excluded from all images.

112   Thank you for suggestion. During the data collection stage, orthoptists (non-ophthalmologist)

113   excluded images with diabetic retinopathy, glaucoma, dense vitreous hemorrhage, fundus

114   hemorrhage, and strong intrinsic vitreous opacity. We do not have the information on the

115   number of images excluded at that stage. Ophthalmologists conducted the final checks after

116   data collection, and they excluded one image with glaucoma.

117

118   3.  Table 2; it is unclear what 32:80±7:36 and 13:58:00±3:19:16 actually mean.

119   Thank you for pointing this out. We have fixed the time unit in our revised version.

120   4.  I am not sure why the authors use Optos to detect MH. OCT should be more accurate, easy,

121      and more common.

122   We agree that OCT is more accurate, easy, and common. However, the retinal disease does not

123   only involve the macula, peripheral retinal lesions are also important. If diagnosis using Optos

124   and AI can comprehensively enabled in the future, it will lead to an improvement in the

125   diagnosis rate. By examining the macular hole, we proved that it is possible to diagnose macular

126 disease even with the use of ultra-wide-field fundus ophthalmoscopy.

127

128

129

130 **Reviewer 3 (Anonymous)**

131 Basic reporting

132 In the table, it is not clear what format and units the time is reported in.

133 Thank you for pointing this out. We have now added a time unit, and changed "32∶80±7:36,

134 13:58:00±3:19:16" to "32.80±7.36, 838.00±199.16" in the revised manuscript.

135 The figure legends should allow the figure to be read without referring to the original article –

136 they may need to be made slightly more descriptive.

137 Thank you for your comments. We have added better descriptions.

138 Figure 1: First, each dataset's image was reduced to 256 × 192 and was input into the model.

139 Next, it was passed through all convolution layers and through the entire binding layer, and it

140 was classified into 2 classes.

141 Figure 2: This is the first one out of 100 ROC curves. The average AUC of 100 ROC curves

142 was almost 1, and all ROC curves were similar.

143 Figure 3: The dark blue color shows the point where the deep neural network is paying attention

144 on the macula and from the point of view of an ophthalmologist.

145 I'm not sure ROC curve is necessary or helpful when the AUC is essentially 1.

146 Thank you for suggestion. Reviewer 4 suggested zooming in on the ROC curve figure, perhaps

147 with the x and y axes at 0.5 or something. We tried, but it was rather confusing.

[Figure]

148

149

150  The image preprocessing is not well described. The images appear to have a circular crop

151  applied to the original image – this should be described.

152  Thank you for your comments. We did not pretreat all the images together. However, as stated,

153  we applied the same treatment to all of them:

154 →"The image amplification process comprised modifications of contrast adjustment, γ

155 correction, histogram equalization, noise addition, and inversion. We used training on these

156 learning images to train a deep convolutional neural network (CNN) and constructed a deep

157 learning model."

158 Was any effort made to center the images, align the disc, or flip left/right eyes to make the

159 images appear similar to the CNN?

160 We did not attempt to center the images, align the disc, or flip left/right eyes to make the

161 images appear similar to the CNN.

162

163 Comments for the Author

164 From an image processing perspective, assuming a good quality fundus image, the detection of

165 a macular hole (a small dark circle in a larger fairly homogenous image) is not that complicated,

166 and thus it is not surprising the CNN works as well as it does, but the results are nonetheless

167 impressive.

168

169 From a clinical utility perspective, it is not clear that this is a solution to an existing clinical

170 problem since macular holes always cause visual loss in the stages included in this study, the

171 rationale for creating a screening program to detect them is less compelling. While perhaps not

172 necessary for publication here, it may strengthen the paper to add some discussion as to how

173 such a program might be used in the real world.

174

175

176

177 **Reviewer 4 (Anonymous)**

178 Basic reporting

179 Introduction

180 For me, the intro is far too short and doesn't really describe the problem in enough detail. There

181 is almost no clinical background, and the discussion around deep learning is too brief. I would

182 suggest expanding the Introduction to cover the following topics:

183 - What is a macular hole? How does it appear in a fundus photo vs. OCT?

184 Thank you for suggestions. We have limited our description to avoid extensive explanations

185 for specific differences, in ophthalmic specialty fields. "The development of optical coherence

186 tomography (OCT) and image resolution improvements have facilitated the diagnosis of

187 macular diseases."

188 - What is the prevalence of macular holes? Some statistics might be helpful

189 Thank you for suggestion. We have now added a sentence to this effect: "The age and gender

190 adjusted annual incidences of primary MH have been reported at 7.9 eyes and 7.4 respectively

191 per 100 000 inhabitants, and the male to female ratio at 1:2.2 (Forsaa et al., 2017)."

192 - What are the complications associated with MH, if left undiagnosed?

193 Thank you for question. If the macular hole is not repaired, the visual prognosis is poor. We

194 have not included this in the manuscript because it is obvious in the field of ophthalmology.

195 - Deep learning is not a machine learning algorithm; it's a sub-field of research within ML

196 Thank you for pointing this out. We have modified our statement from "a machine learning

197 algorithm" to "a sub-field of machine learning algorithm studies".

198 - You state that DL is good generally, but you should give details of why DL is a good approach

199 specifically to your problem. Have other methods been tried previously for MH? Are they

200 inadequate?

201 You have raised an important point. In our previous research we proved that SVM (support

202 vector machine) is inferior to Neural Network (Ohsugi et al., 2017); therefore, this time we did

203 not assess the performance of SVM. Instead, we compared the performance of DL to diagnose

204 MHs with that of human ophthalmologists.

205 - Please cite some other recent DL papers in the context of ophthalmology, especially this one:

206 https://www.nature.com/articles/s41551-018-0195-0

207 Thank you for suggestion. We have now added a citation to reference "(…Ryan et al., 2018)."

208

209 Methods

210

211 - It probably makes more sense to describe the FC dropout layer in the section "Deep learning

212 model", rather than the "Training…" section.

213 Thank you for suggestion. We have made these changes to the revised manuscript:

214 "We performed dropout processing to mask the first total tie layer (FC1), with 50% probability."

215 "Training the deep convolutional neural network" section to "Deep learning model" section.

216 - Lines 132-135 do not make sense - please revise these sentences to be more clear

217 Thank you for observation. We have added "The time required by the ophthalmologists was

218 set as the time taken to complete all answers in the Excel file. The time required for the deep

219 neural network was measured by the internal clock of the computer."

220 - A citation is needed for the Grad-CAM method on line 141

221 Thank you for your suggestion. We have now cited a paper "(…Ryan et al., 2018)."

222 - What cost function did you use? Cross-entropy, or something else?

223 Thank you for your question. We used binary-cross-entropy.

224

225 Results

226

227 I am quite confused about the methods on page 9. Specifically, how (or why) is "deep-learning

228 response time" calculated by the ophthalmologists (line 125)? The description on line 132-135

about data entry is also unclear, particularly the sentence: "In deep learning, a series of tasks was performed for all presented numbers as follows…". My best guess is that the authors are trying to fairly compare the DL computation time with the ophthalmologists' time taken to record the same information. Please revise this section to be more clear.

Thank you for comment. We have clarified the issue in the revised manuscript "The time required by the ophthalmologists was set as the time taken to complete all answers in the Excel file. The time required for the deep neural network was measured by the internal clock of the computer."

Regarding the figures, I think there a few things that can be improved:

- In my opinion, the legends are too short. I personally try to provide enough information in the figures so that a reader could get the gist of the whole paper by reading the legends alone.

Thank you for suggestion. We have added more information to the legends for clarity.

Figure 1: First, each dataset's image was reduced to 256 × 192 and was input into the model. Next, it was passed through all convolution layers and through the entire binding layer, and it was classified into 2 classes.

Figure 2: This is the first one out of 100 ROC curves. The average AUC of 100 ROC curves was almost 1, and all ROC curves were similar.

Figure 3: The dark blue color shows the point where the deep neural network is paying attention on the macula and from the same point of view of an ophthalmologist.

- For Figure 2, I'd suggest zooming in on the ROC curve figure, perhaps with the x and y axes at 0.5 or something. You really can't make anything out otherwise. I'd suggest also including curves from several runs - perhaps the best, worst and average? It'll give readers a better sense of the variability.

Thank you for suggestion. We have set the y coordinate to 0.8-1.

[Figure]

254

[Figure]

255

256 - Figure 3 showing the heat map is not all that informative without a color bar. It also might be

257 useful to include a few examples rather than just one.

258 Thank you. We tried creating a color bar, but it was rather confusing.

259 - Table 2: "Accuracy" is a better term for "correct answer rate". Also please state the unit of

260   measurement time.

261   We have changed the expression "correct answer rate" to "Accuracy" as recommended.

262

263   Other points:

264   - What is the "first" curve? The first experiment you ran? Why not the best curve?

265   You have raised an important question. The data evaluation was accomplished using 90% of

266   images and excluding 10% randomly in 100 ways; therefore, using the best results introduces

267   arbitrariness, so we chose the first one. However, since the AUC is 0.9993 on average and it is

268   nearly 1, we think that any approach is valid in this specific case.

269   - On line 162, is this 13 minutes per image?

270   Thank you for question. We have changed the sentence in the revised manuscript;

271   "Ophthalmologists carried out the test, and the mean (standard deviation) required time was

272   838.00 seconds (±199.16), the mean (standard deviation) accuracy rate was 80.6% (5.9%),

273   sensitivity was 65.9% (15.7%), and specificity was 95.2% (4.3%). The same test was carried

274   out with the CNN model, and the mean (standard deviation) required time was 32.8 seconds

275   (±7.36), and accuracy rate, sensitivity, and specificity were all 100% (Table 2)."

276

277   Conclusions

278   - There's no need to repeat that deep learning is an ML technology.

279   Thank you for suggestion. We have deleted "which is a machine-learning technology."

280   - What are you going to do next?

281   Thank you for suggestion. We have added "Further research with increasing number of sheets,

282   deepening the Layer structure, and using metastasis learning are necessary to confirm our

283   results."

284

285    Grammar, spelling and formatting

286    Overall the language is very good, though there are a few spelling/grammatical errors:

287    - Missing space after "macular holes" (line 19)

288    Thank you for your advice, we changed "macular holes(MHs)" to "macular holes (MHs)"

289    - Optus → Optos (line 61)

290    Thank you for your advice, we changed "Optus" to "Optos."

291    - Lots of unnecessary hyphens in the terms deep-learning and machine-learning (line 71, 88 and

292    various other places)

293    Thank you for your advice, we erased the unnecessary hyphens.

294    - "...using a CNN" (line 88)

295    Thank you for your advice, we changed "using CNN" to "using a CNN"

296    - "The rectified linear unit (ReLU) activation function…" (line 89)

297    Thank you for your advice, we changed the expression "The activation function rectified linear

298    unit (ReLU)" to"The rectified linear unit (ReLU) activation function"

299    - What is meant by a 'tie layer'? Not sure what this means (line 92, 100)

300    Thank you for your advice, we have modified our sentences;

301    "two layers of the total tie layer called fully connection layer (FC 1, 2) were arranged" now

302    reads "the two fully connected layers (FC 1, 2) were arranged".

303    - "The network weights were optimized using stochastic gradient descent (SGD) with

304    momentum…" (line 101-102)

305    Thank you for your advice, we have made the proposed changes.

306    - ROV → ROC, and various grammatical errors afterward (line 111 onwards)

307    Thank you for your advice, we have changed "ROV" to "ROC".

308    - Background data (line 147)

309    Thank you for your advice, we have now changed "Backgrounds data" to "Background data."

310  - Probably better to describe the eye in terms of left/right or OD/OS (line 148)

311   Thank you for your advice, we have changed "side" to "left/right."

312  - Resions → regions (line 201)

313   Thank you for your advice, we have changed "Resions" to "regions."

314

315  Experimental design

316  The research question is not all that well defined in the introduction. Ultimately, the goal was

317  to evaluate the performance of DL algorithm for detecting MH. However, the authors also do a

318  good job of comparing the algorithm to multiple experts; something many papers do not do. I

319  would therefore suggest adding a couple of sentences at the end of the introduction to state that

320  this was also part of the study.

321   Thank you for your suggestion. We added "Deep neural networks have been used to diagnose

322  skin cancer with as much accuracy as that attained by dermatologists (Esteva et al., 2017). We

323  decided to assess the diagnostic capabilities of deep neural networks for macular holes as

324  compared to ophthalmologists' diagnoses."

325   We changed, "in order to determine the accuracy of deep learning for MHs" to "to determine

326  its accuracy based on the ophthalmologists' diagnoses as the gold standards"

327   We added a cited paper;

328  Esteva A, Kuprel B, Novoa RA, Ko J, Swetter SM, Blau HM, Thrun S. (2017) Dermatologist-

329  level classification of skin cancer with deep neural networks. Nature 542:115-118 DOI:

330  10.1038/nature22985.

331

332  My main issue with the overall experimental design relates to how the final model for

333  evaluation was selected. You shouldn't use test accuracy as the basis, but instead use a

334  validation set. Later, in the "Statistical analysis" section, I don't really understand the authors'

335    description of the ROC analysis. I get that there should be one curve per model (100 overall)

336    but I do not understand what is meant by: "We created 100 ROC curves by making 100 patterns,

337    and 10% were thinned out". Some clarification is needed. The authors also state the model was

338    fitted to only 90% of the test data. Presumably this is an error, and the authors mean training

339    data. This would suggest that the authors did indeed use a 10% validation set, but this is unclear.

340    Please revise this section to better describe how the model was tested and evaluated.

341    Thank you for providing these thoughtful comments. We can explain better.

342    First, we divided the image data into 80% of training data and 20% of evaluation data. Next,

343    out of the 20% of the evaluation data, 90% was randomly selected and evaluated in 100 different

344    ways, and 100 ROC curves were derived. Finally, AUC, sensitivity and specificity were

345    calculated for each. We used this method in our past publication (Ohsugi et al., 2017).

[Figure]

346

347

348    Some other points:

349

350    - It's very common to pre-train networks on bigger datasets to boost performance and reduce

351    the time needed to train. Did you try this out?

352    Thank you for suggestion. We did not try this for our study.

353

354    - Images from patients with various complications were excluded. How many? Why?

355    Thank you for questions. During the data collection stage, orthoptists (non-ophthalmologist)

356    excluded images with diabetic retinopathy, glaucoma, vitreous hemorrhage, fundus hemorrhage,

357    strong intrinsic vitreous opacity. We do not have the information on the number of images

358    excluded at that stage. Ophthalmologists conducted the final checks after data collection, and

359    they excluded one image with glaucoma.

360

361    - What was the criteria for "clarity"?

362    Thank you for your question. We comprehensively diagnosed from medical records and OCT,

363    surgical images, etc.

364    - What is "stage 1" MH and why are they not included?

365    Because stage 1 MH has no surgical indications, the number of cases was very small.

366    - You state that 100 models were trained - what was the variability over the 100 runs?

367    Thank you for question. We did not train 100 models. After the learning cycle at the time of

368    model adoption, we considered that no significant changes had occurred.

369

370    Validity of the findings

371    As described above, I am concerned about how the final model performance was evaluated.

372    Running the algorithm 100 times and picking the best one based on testing accuracy is cheating

373    a little bit, although it's not clear from the methods whether this is how it was actually done.

374    Furthermore, given the size of the dataset and near-perfect performance on the test set, I think

375    that cross-validation is necessary to get true sense of algorithm generalizability. I think that

376    given the authors can afford to do 100 repeats on the same data, performing K-fold cross

377    validation should be feasible and would strengthen the reporting.

378    Thank you for your comments. Please refer to the explanations above.

379

380    Given that the authors went to the effort of making an application to capture six experts'

381    gradings, it would be good if the authors could report metrics of interrater agreement (e.g. kappa

382    coefficients). Furthermore, it's important that the authors discuss the limitations of the reference

383    standard used, especially if it was based on the diagnosis of a single grader. This paper gives

384    some good insight into why this is important: https://research.google.com/pubs/pub46802.html

385    Thank you for providing these insights. The purpose of this study was to examine whether DL

386    can distinguish images with MH defined by a retinal specialist from normal images. From this

387    point of view, we assumed that the physician's diagnoses were reliable in this research. To

388    support this premise, we used the additional data such as OCT results and medical records to

389    ensure accuracy of the diagnoses. We used data from six ophthalmologists because we thought

390    that it was needed more than one to obtain a better estimate of the time required for reading an

391    image, and to obtain ophthalmologists' sensitivity and specificity, etc. Please note that we did

392    use data from six ophthalmologists to examine the consistency of their answers. Because of this,

393    we do not consider the kappa coefficient and coincidence rate of the answers of six

394    ophthalmologists as necessary.

395

396    Comments for the Author

397    Overall, the paper is well written and demonstrates that DL is a powerful approach for

398    classifying MH in wide-angle retinal photographs. The choice of metric is appropriate and the

399    comparison with multiple raters is great to see. Some more background information about MH

400    would be useful for readers outside of the discipline, and would help to contextualize why this

401    work is important.

402

403    Ultimately, I am concerned that the results do not reflect the realistic real-world performance

404    of the proposed method. With further clarification of how the model was tested and evaluated,

405    I think that most of concerns will be addressed. However, I would maintain that cross-validation

406    would be a more appropriate way of assessing the generalizabiltiy of the CNN.

---

## Round 0.3 · accepted · Accept

Thanks for addressing most of the previously raised issues.